# Towards a new approach to reveal dynamical organization of the brain using topological data analysis

Manish Saggar[1], Olaf Sporns[2], Javier Gonzalez-Castillo[3], Peter A. Bandettini[3,4], Gunnar Carlsson[5,6], Gary Glover[7] & Allan L. Reiss[1,7,8]

Little is known about how our brains dynamically adapt for efficient functioning. Most previous work has focused on analyzing changes in co-fluctuations between a set of brain regions over several temporal segments of the data. We argue that by collapsing data in space or time, we stand to lose useful information about the brain's dynamical organization. Here we use Topological Data Analysis to reveal the overall organization of whole-brain activity maps at a single-participant level—as an interactive representation—without arbitrarily collapsing data in space or time. Using existing multitask fMRI datasets, with the known ground truth about the timing of transitions from one task-block to next, our approach tracks both within- and between-task transitions at a much faster time scale (~4–9 s) than before. The individual differences in the revealed dynamical organization predict task performance. In summary, our approach distills complex brain dynamics into interactive and behaviorally relevant representations.

[1] Department of Psychiatry & Behavioral Sciences, Stanford University, 401 Quarry Rd, St 1356, Stanford, CA 94305, USA. [2] Department of Psychological & Brain Sciences & Network Science Institute, Indiana University, 1101 East 10th Street, Bloomington, IN 47405, USA. [3] Section on Functional Imaging Methods, National Institute of Mental Health, NIH, Building 10, Room 1D80, 10 Center Dr. MSC 1148, Bethesda, MD 20892, USA. [4] Functional MRI Core Facility, National Institute of Mental Health, Building 10, Room 1D80B, 10 Center Dr. MSC 1148, Bethesda, MD 20892, USA. [5] Department of Mathematics, Stanford University, 383L, Third Floor, Building 380, Stanford, CA 94305, USA. [6] Ayasdi, Inc, 4400 Bohannon Drive, Suite 200, Menlo Park, CA 94025, USA. [7] Department of Radiology, Stanford University, Lucas Center P-074, Stanford, CA 94305, USA. [8] Department of Pediatrics, Stanford University, 725 Welch Road, Palo Alto, CA 94304, USA. Correspondence and requests for materials should be addressed to M.S. (email: saggar@stanford.edu)

Understanding how our brain dynamically adapts from one task to the next is vital for comprehending typical and atypical brain functioning. With the advent of modern neuroimaging modalities, sophisticated attempts have been made to employ time-evolving estimates of inter-regional functional connectivity (FC), during both at rest[1] and evoked[2,3] experimental paradigms, to study how the brain functionally reconfigures at the scale of seconds to minutes[4]. Most existing analytical methods collapse data both in space and time at the onset of the analysis[1,3], thereby decreasing the spatiotemporal scale of the data beyond that set during acquisition, and likely hindering our ability to extract fine-grained information on brain dynamics and transitions.

Different techniques have been proposed to empirically characterize FC over the entire scan time (typically 6–10 min)[5–7]. Using such methods on "resting" data (i.e., in the absence of externally demanding tasks), we have learned that the intrinsic brain activity is organized into several major systems that were previously apparent only during task-induced paradigms[8,9]. However, collapsing FC over the entire scan time neglects the variation in spatiotemporal properties of the neural processes of interest[10,11]. In fact, recent studies show that within-participant properties of FC can vary considerably within the confines of individual scans (on a timescale of several seconds)[10]. These temporal variations have been observed in humans[12–15] and other species[16,17]. Concurrent neuroimaging studies, involving functional magnetic resonance imaging (fMRI) and electrophysiological recordings, have suggested that the origin of temporal variations is neurophysiological[4,14,18], and that such variations contain clinically relevant information[19]. Thus, our current understanding of the brain functioning based on "average" FC, and the accompanying inferences, is at best, incomplete.

To analyze time-varying FC (a.k.a. dynamical FC or dFC), methods based on sliding-window[13,16,20], single-volume co-activation patterns[21], wavelets[13], change-point detection[22], deconvolution[23], multiplication of temporal derivatives[4,24], and temporal Independent Component Analysis[12] have been proposed. While these novel methods provide valuable insights, several fundamental issues remain unresolved, including (1) uncovering the temporal and spatial scales that best capture clinically and behaviorally relevant brain dynamics; (2) understanding whether the dynamical landscape of possible configurations is best conceptualized as continuous or discrete[1]; and (3) recognizing what constitutes healthy and aberrant dynamics. Tackling these issues requires novel tools that avoid arbitrarily collapsing data in time and space early in the analysis, provide interpretable visualizations of how the brain traverses its dynamical landscape—namely data-driven abstractions that attempt to capture the stream of thought originally proposed by William James (Chapter IX)[25]—and permit quantification of these dynamic trajectories in behaviorally and clinically relevant ways that allow comparisons across conditions, participants, and populations.

Here, we present such a novel method to represent brain's overall dynamical organization as a combinatorial object (or graph), without arbitrarily collapsing data in space or time. The proposed representations can be interactively visualized, quantified in a variety of ways using graph theory, and constructed at the level of individual participants, making them suitable for exploratory research and translational purposes. To achieve this goal, we employed and extended a tool from the field of Topological Data Analysis (TDA) called Mapper[26,27]. The intuition behind Mapper is to reduce a high-dimensional data set into a combinatorial object (see Supplementary Fig. 1). Such an object attempts to encapsulate the original shape, or the topological and geometric information, of the data by representing similar points nearby than dissimilar points. Intuitively, such a representation is analogous to generating a topographical map that can capture the essential features of a landscape. Although TDA-based Mapper is in principle similar to other traditional manifold learning (or non-linear dimensionality reduction) algorithms (e.g., ISO-MAP[28]), it provides several advantages over them. For example, unlike manifold learning, Mapper makes fewer assumptions about the underlying data. Further, unlike other methods, Mapper represents the underlying landscape as a graph, which is robust to noise and its properties can be easily estimated for better quantification. Additionally, the coordinate and deformation invariance properties of Mapper make it suitable for examining data across participants and projects[29,30]. The TDA-based Mapper has been previously applied to reveal the shape of genetic data in breast-cancer patients[31], neuronal data from the visual cortex[32], biomolecular folding pathways[33], voting behavior in the U.S. House of Representatives[29], and anatomical data in patients with fragile X syndrome[30].

We tested the efficacy of our approach in an fMRI dataset with known ground truth about the timing of transitions between mental states as dictated by tasks[34]. This dataset, originally acquired to evaluate the behavioral significance of FC states[20], consists of ~25 min continuous scanning sessions, during which participants performed multiple tasks (two sets of working memory, arithmetic operations, and a visuospatial search task) along with resting for short blocks of time (~3 min), to simulate ongoing cognition. Since the transitions between different cognitive processes were experimentally constrained, as opposed to self-directed transitions during rest, the paradigm provides ground truth regarding the timing and nature of cognitive states and their transition[34]. We hypothesized that our approach would (1) provide novel insights about how the brain dynamically adapts in a multitask paradigm at the level of single individual; (2) capture dynamical transitions in neural processes at higher temporal resolution than before; and (3) provide neural markers for individual differences in task performance.

We performed rigorous reliability and validation analysis for the proposed approach, including comparison with three different null models, replicating our results in an independent dataset from the Human Connectome Project[35], and a thorough parameter perturbation analysis. To derive novel biological insights about brain functioning and its dynamics, we also provide methods to quantify and visualize the generated representation and anchor the representation and its features into neurophysiology by revealing pseudo-instantaneous whole-brain activation patterns.

Using multiple fMRI datasets, our approach tracks both within- and between-task transitions at a much faster time scale (~4–9 s) than before. Further, the individual differences in the organization of whole-brain activity maps predict task performance. Altogether, we provide a novel method to distill complex brain dynamics associated with ongoing cognition into a set of interactive and behaviorally relevant representations by taking full advantage of the original temporal and spatial scales of the data.

## Results

**Revealing dynamical organization of the brain**. To test the efficacy of our approach in estimating a representation of the brain's dynamical organization and in capturing transitions in the whole-brain activity, we employed already collected fMRI data from Gonzalez-Castillo et al.[34]. These data were collected while participants performed in a Continuous Multitask Paradigm (CMP) with experimentally constrained transitions from one task to the next.

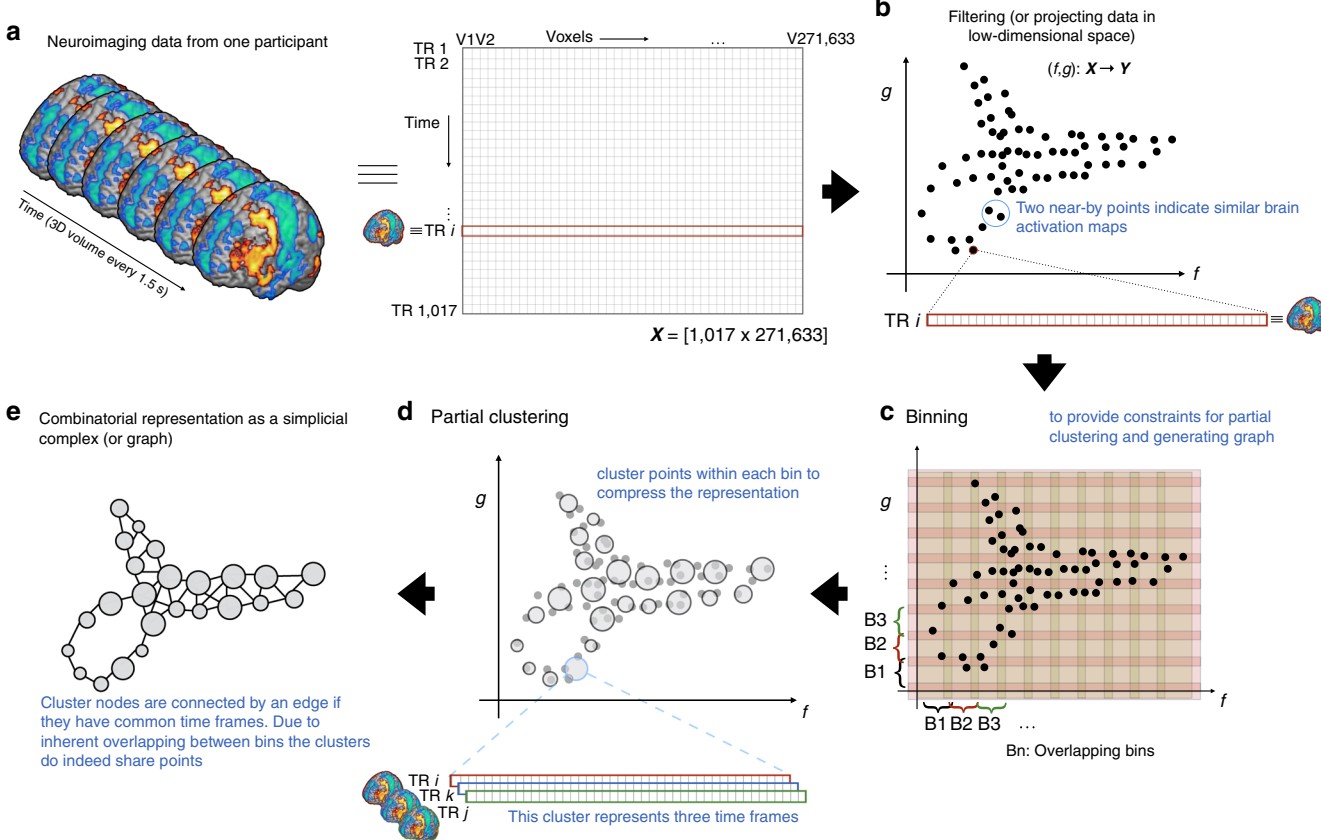

**Fig. 1** Application of Mapper to 4D fMRI data. **a** Pre-processed four-dimensional fMRI data from each participant was fed into the analysis. For each participant, the entire data matrix (i.e., #TRs × #Voxels) was used for analysis. **b** Non-linear dimensionality reduction was done during the filtering step to project fMRI data into a lower two-dimension set (represented by dimensions (f, g)). **c** Two-dimensional binning was then performed by dividing the lower-dimension space into smaller bins (determined by the resolution parameter, R) with certain overlap (determined by the gain parameter, G). **d** Partial clustering was then performed to get a compressed representation by collapsing data into fewer nodes, where each node represents a cluster, and the size of each node depicts the number of data points inside each cluster. **e** After clustering, nodes that share data points (i.e., time frames in this case) are linked together with an edge to create the final compressed combinatorial representation (or graph)

After standard fMRI preprocessing, each participant's 4D fMRI data were first transformed into a matrix, such that the rows corresponded to the individual time frames (or volumes) and the columns corresponded to the intensity value at each voxel. Thus, each row of this matrix represents the entire brain volume at any time point during the session (Fig. 1a). The TDA-based Mapper is next employed on this matrix to perform four steps—filtering, binning, partial clustering, and finally constructing the shape graph (Fig. 1b−e). Although data-driven optimization was employed to find the best set of parameters for each of the Mapper steps, we observed that the presented results are stable in the face of extensive parameter perturbations (see Reliability of shape graphs).

As a first Mapper step, we applied a filtering (or dimensionality reduction). This step is similar to the standard dimensionality reduction techniques used in the machine learning literature. However, unlike traditional linear dimensionality reduction techniques, like principal component analysis (PCA) or multi-dimensional scaling (MDS), we employed a nonlinear dimensionality reduction method using a variant of stochastic neighborhood estimation (SNE[36,37]). Nonlinear methods like SNE allows for preservation of the local structure in the original high-dimensional space after projection into the low-dimensional space, which is typically not possible with linear methods like PCA or MDS[36]. Thus, the time frames with similar activation patterns in the original high-dimensional space will be projected closer to each other in the reduced dimensional space (Fig. 1b).

To encapsulate the low-dimensional representation generated by the filtering step, Mapper employs binning (or partitioning) (Fig. 1c), followed by partial clustering within each bin. The binning step partitions the low-dimensional space into over-lapping bins by using two parameters—number of bins (or resolution (R)) and percentage of overlap between bins (or gain (G)). Within each bin, single-linkage clustering is performed to condense the time frames into a set of one or more clusters (Fig. 1d). This step results in a compressed representation, as fewer points (or clusters) are now required to represent the data as compared to the entire set of time frames. On average, for each participant, the compressed representation contained ~279 points (SD = 60) (as opposed to 1017 acquired time frames). It is important to note that this clustering step is different from traditional temporal smoothing as the time frames within each cluster are not averaged and the mapping between individual clusters and their corresponding time frames is preserved.

Finally, to generate a combinatorial object or shape graph from the low-dimensional compressed representation, the Mapper treats each cluster as a node in the graph and connects these nodes with an edge if they share time frames (Fig. 1e). The final shape graph can be conceptualized as a low-dimensional depiction of how the brain dynamically evolved across different functional configurations during the scan. While the actual interpretation of the latent variables associated with the projected

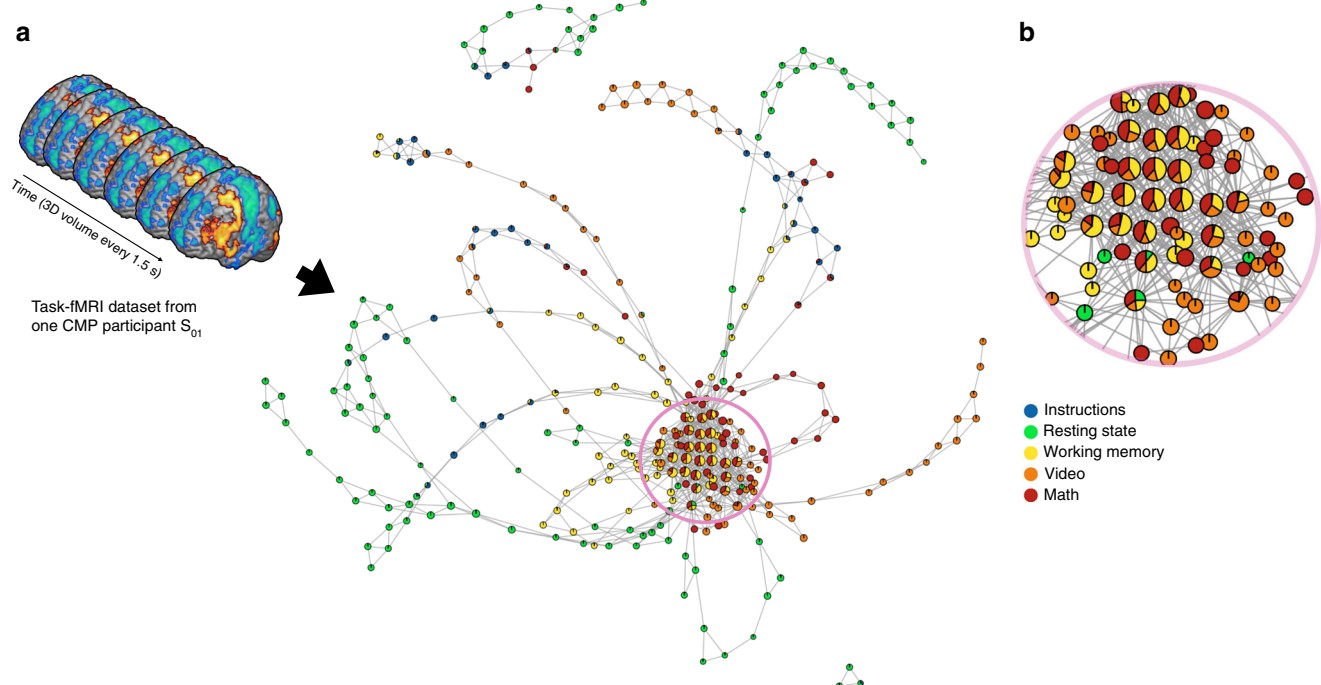

**Fig. 2** Revealing the shape of brain's dynamical organization. **a** Depicts the shape graph for one of the representative participants ($S_{01}$) from the CMP dataset. The shape graph was annotated using colors and pie-chart visualization scheme to depict how the tasks were represented in each shape graph. **b** Shows a zoomed-in version of the densely connected region of the shape graph to show the use of pie-chart visualization

low-dimensional space may differ across participants, the topological relationships encoded by the shape graph itself are interpretable and comparable across participants (as we shall discuss later).

To reveal the underlying temporal structure in the CMP dataset, the aforementioned Mapper approach was applied to each participant in the CMP dataset. For qualitative analysis, we annotated the nodes in these shape graphs with colors based on the corresponding task at each time frame (Fig. 2a). Further, if a node contained time frames from multiple tasks, we visualized that node using a pie chart denoting the proportion of time frames that belong to each task within such node (Fig. 2b). Graph theoretical metrics were next used to quantify the topological properties of each participant's shape graph.

**Quantifying the mesoscale structure of shape graphs.** Graph theory (or Network Science) is currently widely used in the field of neuroscience to provide summary statistics of the complex interactions between different entities or nodes. While interesting insights can be captured by analyzing properties of each node or edge in the network (i.e., at the local scale) or by analyzing the network as whole (i.e., at the global scale), the intermediate (or mesoscale) properties appear particularly well suited for analyzing and comparing the structure of complex networks[38,39]. In particular, considerable effort has gone into identifying two distinct types of mesoscale structures in a variety of complex networks. The first and perhaps the most widely used mesoscale structure is the community structure, where cohesive groups called communities consist of nodes that are densely connected to other nodes within communities while being only sparsely connected to nodes between communities[40]. In the context of shape graphs, the presence of communities could represent a modular organization with specialized whole-brain functional configurations for different types of information processing (or tasks). An increasingly second most typical mesoscale structure is the core−periphery

structure[41]. Here, one attempts to determine the core nodes, which are not only densely connected to each other but are also central to the entire network. A presence of core nodes in shape graphs could indicate whole-brain functional configurations that consistently occur throughout the scan. For example, core nodes could represent neural processes related to task-switching that the brain consistently passes through during a multitask experimental paradigm. The peripheral nodes on the other hand are only sparsely connected. The examination of the core−periphery structure of a graph could reveal the overall arrangement of the network[39]. It is important to note that in the real world, networks can have both communities and core−periphery structures and hence it is desirable to investigate both simultaneously.

To exemplify how topological properties of shape graphs can provide behaviorally relevant information at the single-participant level, we estimated both community and core−periphery mesoscale structures. Briefly, to estimate the degree or quality of the community structure in shape graphs, we assessed the widely used quality function $Q_{mod}$[42]. The community assignment for each node in a shape graph was chosen to be one of the four tasks (i.e., Rest, Memory, Video, and Math) based on the majority of time frames contained in the node that belonged to the respective task. Across the CMP dataset, we observed participants' shape graphs with varying degree of modularity (ranging from $Q_{mod} = 0.37$ to $0.61$ with a mean $= 0.48$ and SD $= 0.07$; Fig. 3a). Remarkably, the degree of modularity was observed to be associated with task performance across the three CMP tasks (%correct $r = 0.56$, $p = 0.016$; Fig. 3b), such that high modularity was associated with better performance on the CMP tasks. Thus, highlighting that participants with a higher degree of community structure in their shape graph better performed across different cognitive tasks during the CMP. In other words, participants with specialized whole-brain configurations for different tasks were those with the highest overall task performance. We examined this claim using an independent validation analysis (see Anchoring topology of shape graphs into

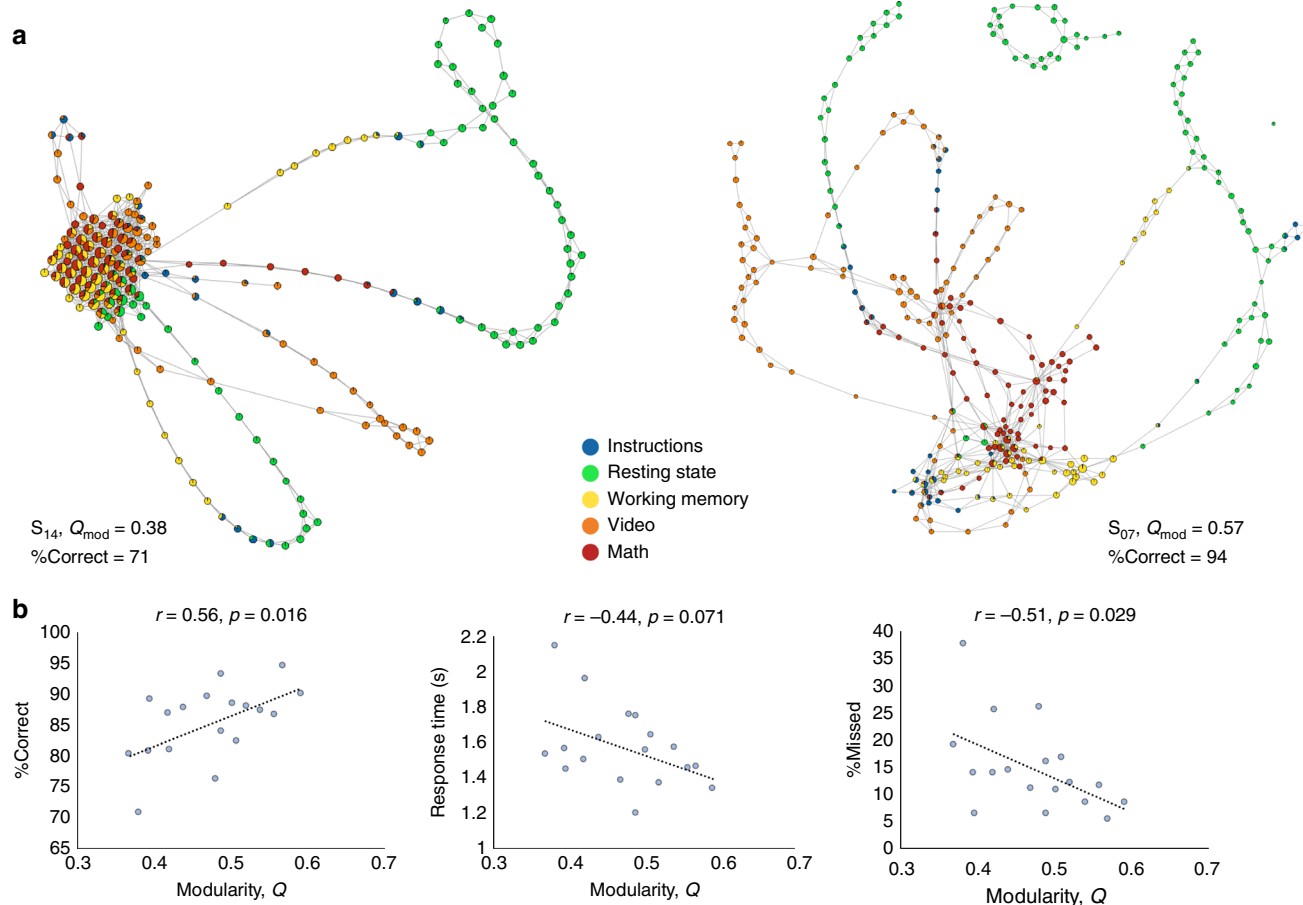

**Fig. 3** Quantifying the mesoscale community structure of shape graphs. **a** Shows two shape graphs from two representative participants ($S_{14}$ and $S_{07}$). As depicted, $S_{14}$ had a low modularity score, i.e., nodes with different task types are connected to each other without any preference for same task nodes. On the other hand, shape graph from $S_{07}$ depicts high modularity structure with nodes preferentially connected to other nodes of same task type. **b** The modularity score ($Q_{mod}$) was observed to be associated with task performance, such that higher modularity in the shape graph was associated with better task performance

anatomy). Please note that because the quantification of community structure was done on the shape graph as a whole (i.e., across tasks), we combined task performance measures across the working memory, video and math tasks.

To quantify the core−periphery structure in each participant's shape graph, we employed the generalized Borgatti and Everett[41] algorithm that provides a coreness score (CS) for each node. This algorithm assigns CS along a continuous spectrum with nodes that lie most deeply in a network core with a CS ~1 to those that are in the periphery with a CS ~0[39]. Figure 4a presents shape graphs annotated (or colored) by the task type as well as CS for two representative participants. Remarkably, across all participants, the nodes containing resting state time frames were mostly represented in the peripheries (mean $CS^{Rest}$ [SD] = 0.15 [0.06]), while the nodes containing time frames from cognitively demanding tasks mainly lied relatively deeper inside the shape graph (mean $CS^{W.M.}$ [SD] = 0.28 [0.04]; mean $CS^{Math}$ [SD] = 0.30 [0.04]; and mean $CS^{Video}$ [SD] = 0.22 [0.06]). One-way ANOVA revealed a significant effect of the task ($F(3,51) = 24.06$, $p < 0.0001$), such that $CS^{Rest}$ was observed to be significantly lower than the CS of other three tasks, while coreness scores for the working memory and math tasks were similar but higher than that of the video task. This result indicates more consistency in the whole-brain functional configurations was present during math and memory task as compared to the less demanding resting state.

To test the validity of the observed non-trivial arrangement of resting state nodes in the peripheries while the cognitively demanding nodes in the core, we employed three different null models. Overall, the CS of nodes with resting time frames was observed to be lower than all the three corresponding null models ($ps < 0.001$), while the CS of nodes with working memory or math frames was higher than all the three corresponding null models ($ps < 0.005$). No significant difference was observed for the CS of nodes with time frames from the video task and the corresponding null models (Fig. 4b).

**Anchoring topology of shape graphs in anatomy**. To ground the shape graphs and their properties into neurophysiology, we provide three approaches that attempt to reveal the underlying patterns of brain activity putatively responsible for the observed topological features. In the first approach, we use spatial mixture modeling (SMM)[43] to reveal changes in brain activation maps from one time frame to the next. The SMM approach includes fitting a mixture of distributions and using a spatial Markov random field to regularize the labeling of voxels into null, activated or deactivated. Thus, for each node in the shape graph and the containing time frames, we generated whole-brain activation (and deactivation) maps. To interactively examine the temporal variations in these activation maps, we developed a web-tool (Supplementary Figs. 2−3 and Supplementary Movie 1). The

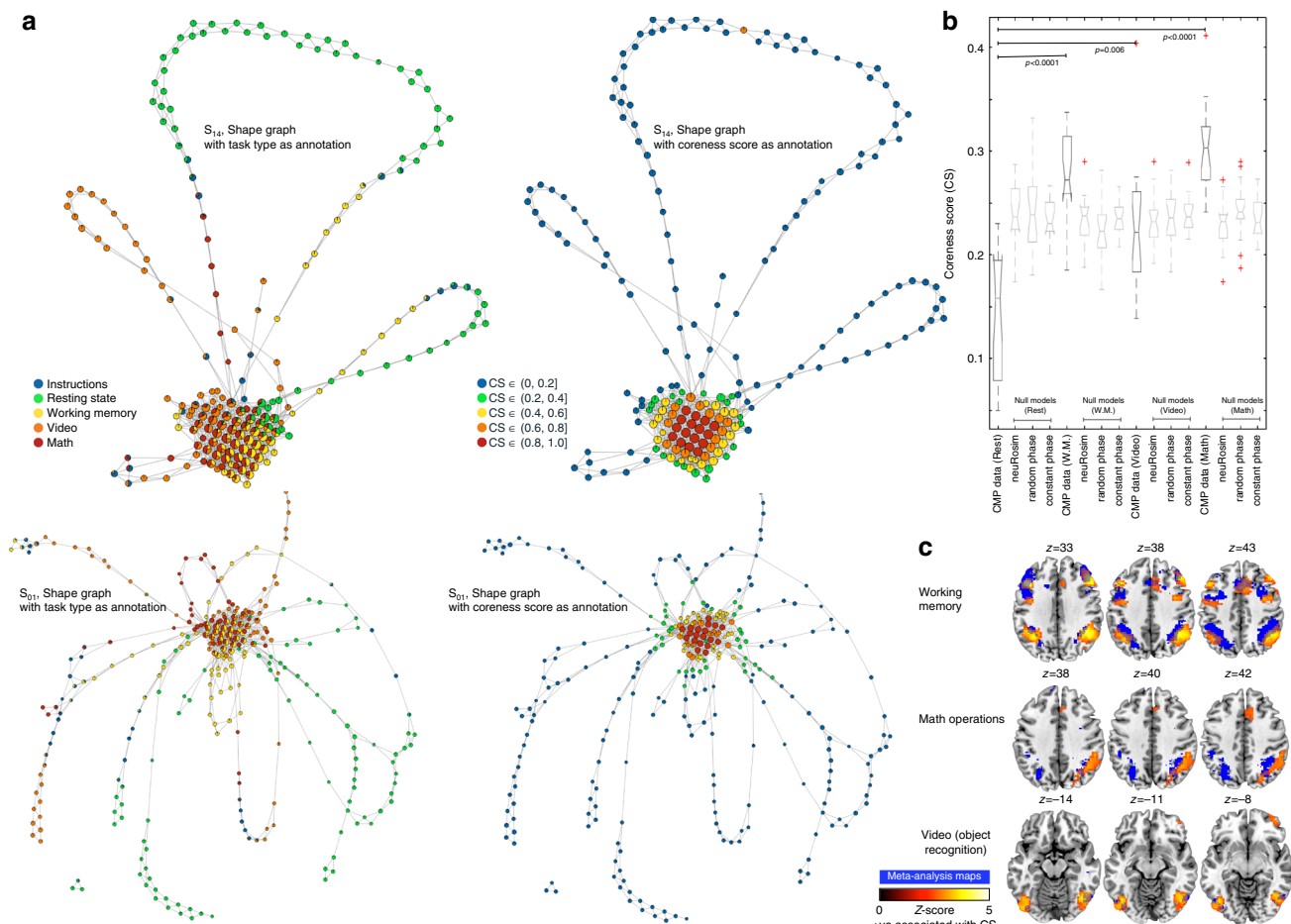

**Fig. 4** Quantifying the mesoscale core—periphery structure of shape graphs. **a** Shows shape graphs for two representative participants ($S_{14}$ and $S_{01}$). In the left column, we annotated the nodes using task-type and in the right column the nodes are annotated using the coreness scores (CS). As evident, the more central and densely connected nodes have high CS and more sparsely and peripheral nodes have less CS. **b** Shows boxplots of CS derived from the CMP data as well as from the three null models (neuRosim, phase randomization with random, and constant phase). Across the four CMP tasks, nodes containing resting time frames had the lowest CS, while the nodes containing time frames from cognitively demanding tasks mainly lied relatively deeper inside the shape graph (i.e. higher CS). The CS of nodes with resting time frames was also lower than all the three corresponding null models ($ps < 0.001$), and the CS of nodes with working memory or math frames was higher than all the three corresponding null models ($ps < 0.005$). No significant difference was observed for the CS of nodes with time frames from the video task and the corresponding null models. **c** Shows the brain regions that were positively associated with coreness scores, i.e., higher the CS, higher the activation in these regions. The group-maps (in red-yellow scale) for positive relation with CS are overlaid on the task-associated meta-analysis maps from NeuroSynth[44]. As shown, higher CS was associated with higher engagement of task-related brain regions

web-tool allows for a better interpretation of the shape graph and its topological properties (i.e., edges and nodes). For example, in real time, a user can move the Time-Frame slider (across time frames) to simultaneously highlight respective nodes in the shape graph; see transitions in corresponding whole-brain activation maps; and observe correlations of the activation maps with known large-scale brain networks[9] (Supplementary Movie 1). Thus, allowing an inspection of neurophysiology at the whole-brain level and the highest temporal resolution (limited only by acquisition rate). When using this tool on the CMP dataset, we can observe how nodes associated with the memory task correspond to whole-brain activation maps with significant activity in the dorsolateral prefrontal cortex, and visual cortex; while for the math task we see the engagement of parietal regions previously associated with arithmetic processing (Supplementary Fig. 4). These results validate the cognitive relevance of the activity maps generated from the shape graphs as they identify regions commonly activated by these tasks.

In the second approach, to anchor the overall topological properties of the shape graph into neurophysiology, we utilized the traditional group-based generalized linear model (GLM) analysis. Specifically, we examined the neurophysiological basis for the observed non-trivial mesoscale structure of core–periphery in the shape graphs. For this analysis, the coreness score of each node was mapped back to the individual time frames contained in that node. Thus, if a node has a CS of 0.5, then the time frames contained in that node also received a CS of 0.5. Using multiple regression, the CS for each time frame was entered for each task creating four explanatory variables. Two contrasts were run to examine brain regions that show positive as well as negative association with the coreness scores. The cluster-corrected ($Z > 2.3$ and FWER $p < 0.05$) group-level results are shown in Fig. 4c and Supplementary Table 1. For the positive association contrast, during the working memory task, higher coreness scores were associated with increased engagement of the bilateral dorsolateral prefrontal cortex (DLPFC), bilateral insula

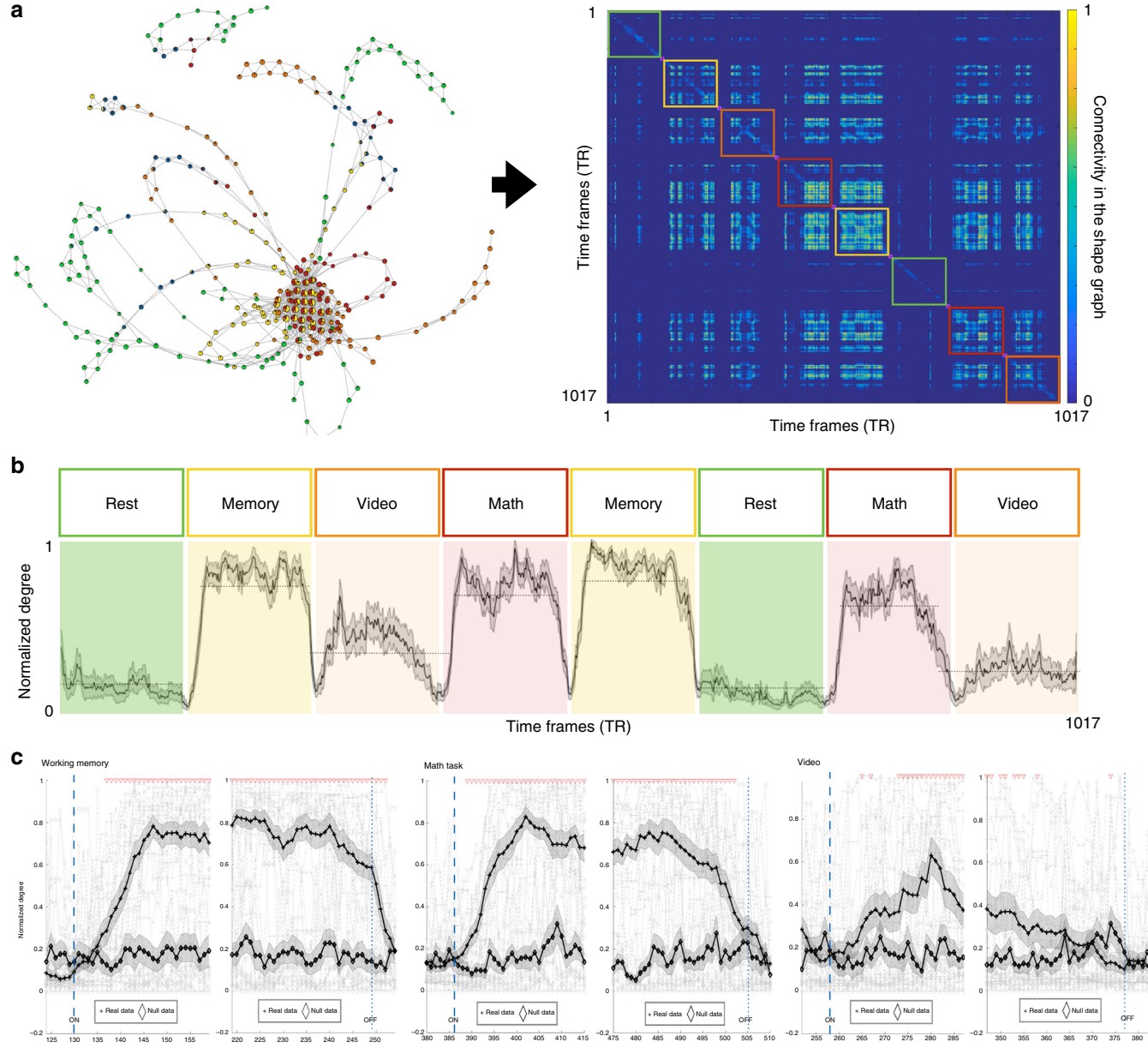

**Fig. 5** Capturing temporal transitions at the level individual time frames. **a** Shows a temporal connectivity matrix (TCM) for participant $S_{01}$ from the CMP dataset. A TCM (size: #time frames × #time frames) shows how each time frame is connected (or similar) to all other time frames. The time frames are connected (i.e. non-zero value in the TCM) if they share a node in the shape graph or if the nodes containing these time frames are connected by an edge in the shape graph. We have manually highlighted the task blocks on the TCM to show the task-related modular structure. **b** Shows that the degree of TCM nodes can capture the transition between tasks. Here we show average degree across all 18 participants as a solid-line, shaded region shows the standard error around the mean. The dotted lines denote detected change points, which mark the transitions for the eight task blocks. **c** Shows the effectiveness of capturing temporal transitions extracted from the CMP data as compared to transitions extracted from the phase-randomization null model. The blue dashed line denotes onset of task block and the blue dotted line indicates offset of task block. As compared to the null model data (denoted using diamonds), a time frame by time frame analysis revealed that the degree of TCM generated using real data (denoted using + signs) could capture both the onset and offset of each task within a matter of a few time frames. The red inverted triangles on top denote point-by-point significant difference between transitions from real as compared to null data

and lateral occipital cortex, and paracingulate gyrus. Higher coreness scores during the math task were associated with increased engagement of the R. angular gyrus, inferior parietal sulcus areas and the paracingulate gyrus. Lastly, for the video task, higher coreness scores were positively associated with activation in the bilateral fusiform gyrus and right frontal pole. Qualitatively, the brain regions associated positively with coreness scores largely overlapped with regions previously shown to be recruited for the respective tasks (Fig. 4c). This qualitative

assessment was performed by overlaying the observed results on the meta-analysis maps generated by NeuroSynth.org[44] for each task term (i.e., "working memory," "arithmetic," and "object recognition," respectively). For the negative association contrast, across all three tasks, significant clusters were observed in the posterior cingulate cortex (PCC) and medial prefrontal cortex (Supplementary Fig. 5), suggesting nodes with lower coreness scores (i.e., periphery nodes) were associated with increased activation in the PCC irrespective of the task type. No significant

cluster was observed for positive or negative association with coreness scores in the resting state task. Overall, these results suggest that core nodes in the shape graph represent task-related activation and putatively associated cognitive effort, whereas sparsely and peripherally connected nodes in the shape graph represent task-unrelated activation presumably related to task-negative default mode regions.

Lastly, an independent whole-brain FC analysis was run to validate the TDA-derived finding that individuals with higher task performance putatively evoked task-specific brain configurations as compared to individuals with lower task performance. Two groups were formed using a median split based on the overall task performance (low ($n = 9$) and high performers ($n = 9$) with a median split at 86.9% accuracy). The whole-brain FC was estimated by sampling data from a set of 264 brain regions to make inferences at the regional and systems level (Supplementary Fig. 12). We used an independent and well-established brain parcellation scheme that was previously identified using a combination of resting-state FC parcellation as well as task neuroimaging meta-analysis[45]. The FC between each pair of brain regions was estimated using Pearson correlations ($r$), and was converted using Fisher's z-transform for further analysis. After estimating whole-brain FC matrices for each task-block, we calculated the similarity between FC matrices derived from different task blocks within each participant. To perform a group-level comparison, the average similarity between FC matrices across tasks, for each participant, was estimated and a two-sample $t$ test was run to compare the low-performance group from the high-performance group. A significant $t$-statistic ($t(16) = 2.50$, $p = 0.0236$) was observed, such that participants in the low-performance group (compared to the high-performance group) had higher average similarity between FC matrices across tasks (Supplementary Fig. 12B–C). Thus, validating the TDA-derived prediction that participants who performed better across tasks evoked task-specific brain configurations.

**Capturing temporal transitions in brain dynamics**. To estimate the temporal transitions in the whole-brain activity maps, we converted the TDA-generated shape graph into an adjacency matrix in the temporal domain (i.e., a temporal connectivity matrix (TCM)). It is important to note that a TCM is representing similarity (or connectivity) in time and not in space like the standard FC matrices that represent anatomical region-by-region connectivity. Here, the time frames are considered connected if they share a node or if the nodes containing these time frames are connected by an edge in the shape graph. As evident from Fig. 5a, for a representative participant, the TCM was observed to be modularly organized, with densely connected frames within each task block and across blocks of the same task.

Remarkably, by directly estimating the degree (or the total number of connections) at each time frame in the TCM, we could capture the transitions between and within task types at the level of a few time frames. Inherently, a higher degree at any time frame implies greater similarity of that frame with other frames. Thus, during the task blocks, other than resting state block, the evoked activity associated with the stimuli/task would cause the time frames to be highly coherent or similar within each block and across the repetition of the same task (and hence more connected in the TCM), thereby leading to a higher degree value. During the resting state blocks (as well as during between-task instruction periods) the brain activation patterns were driven by intrinsic (and not evoked) activity, which would lead to less coherent or dissimilar patterns and hence a lower degree value. Thus, a task-switch from an evoked task to an instruction period or vice versa would lead to a change in degree values at the level

of a few time frames (Fig. 5b, c). As shown in Fig. 5b, the average normalized degree of TCM reveals the between task transitions. Using a standard change point detection algorithm[46], we were able to retrieve eight transitions in the mean normalized degree, corresponding to the eight task blocks. Additionally, for each task type, we quantified how quickly the degree of TCM can capture both onsets and offsets of each task block. As compared to the null model generated using a phase randomization technique (see Validation of the shape graphs against null models), a time frame by time frame analysis revealed that the degree of TCM generated using real data could capture both the onset and offset of tasks within a matter of a few time frames (Fig. 5c).

Similar to the coreness score analysis, a one-way ANOVA revealed a significant effect of the task for the degree at each time frame in the TCM ($F(4,68) = 104.27$, $p < 0.0001$). The working memory task had the highest degree (mean $d_{TCM}^{WM} = 0.71$ [0.11]), followed by math (mean $d_{TCM}^{Math} = 0.60$ [0.13]), then video (mean $d_{TCM}^{Video} = 0.31$ [0.17]) and lastly, resting state (mean $d_{TCM}^{Rest} = 0.15$ [0.06]).

**Reliability of shape graphs**. We performed three reliability analyses. First, for each participant, we split the data into two halves and ran our approach on each half independently. We hypothesized that if our approach is reliable, we should observe similar shape graphs and their properties for the two halves. As hypothesized, the independently generated shape graphs for both halves had similar mesoscale structural as well as temporal properties as observed with the full dataset. In the shape graph from each half of the data, we found: (1) high modularity associated with better performance across tasks; (2) core−periphery structure, such that coreness scores for resting state was lower than cognitively demanding tasks; and (3) between-task temporal transitions at the level of few time frames (Supplementary Fig. 6). These findings suggest that our approach is replicable even when only a part of the data is used to estimate shape graphs.

As a second attempt to estimate reliability, we replicated our approach on an entirely different dataset with distinctive data acquisition parameters, preprocessing steps and experimental design. We used fMRI dataset from 38 unrelated participants from the Human Connectome Project (HCP)[34] who performed a working memory task. The working memory task consisted of two runs (~5 min each) and three conditions (2-back, 0-back, and fixation). As compared to the CMP dataset, the HCP dataset had only one task for each session. Thus, instead of examining dynamical organization across tasks, we examined the organization within a single task (but across conditions). Using similar Mapper parameters as used in the CMP dataset, the shape graphs generated from the HCP data also contained: (1) a hybrid mesoscale structures of modularity as well as core/periphery; (2) a similar pattern of significantly less coreness scores for fixation (or resting) nodes as compared to cognitively demanding working memory nodes; and (3) temporal transitions extracted from TCM showing "within-task" transitions from one condition to the next (Supplementary Fig. 7).

As a third measure towards estimating the reliability of our approach, we tested the effect of parameter perturbation on shape properties and their relation to behavioral task performance. We varied both the TDA parameters—i.e., number of bins (or resolution, $R$) and percentage of overlap between bins (or gain, $G$)—to generate 49 different variations. Results are shown in Supplementary Fig. 8. Overall, the shape properties (e.g., the core-periphery structure) were reliably observed in a majority of parameter variations. Further, the association between modularity and task performance (in the CMP data) was also reliably significant across a majority of parameter variations. Altogether

indicating robustness of results in the face of parameter perturbation.

**Validation of the shape graphs against null models**. To benchmark and validate the results generated from our approach, we employed three different null models. The first null model was designed to test whether the metrics generated from our approach (including the shape graph) were mainly driven by non-brain physiological signals (e.g., cardiac and respiratory) and spatiotemporal noise. In addition to these primary sources of noise, the individual variations in neuroanatomy were also included in the model, by using each anatomical scan (T1-weighted scan) for defining the baseline. The second and third null models were designed to test whether the task-unrelated non-stationarity primarily drove the shape graph and its properties. To generate last two null models, phase randomization (with and without constant phase sequence) of the original dataset was performed independently for each individual. After applying our approach to the data generated from these null models, we did not observe: (1) core−periphery or modularity characteristics in any of the null models, (2) task-dependent variation in the coreness scores, and (3) any structure in the temporal transitions across tasks (Supplementary Fig. 9).

In addition to these null models, we also tested whether head movement artifacts were responsible for individual differences in the TDA-generated shape graphs. No correlation was observed between the shape-graph properties (e.g., coreness score) and measures of head movement artifacts (e.g., mean relative and max absolute head displacement; all $ps > 0.05$).

## Discussion
How our brain dynamically adapts to perform different tasks is vital to understanding the neural basis of cognition. Moreover, the brain's inability to dynamically adjust to environmental demands and brain's aberrant dynamics, on average, have been previously associated with disorders such as schizophrenia[19], bipolar disorder[47], depression[48], and dementia[49]. The high spatiotemporal dimensionality and complexity of neuroimaging data make the study of whole-brain dynamics a challenging endeavor. Researchers and clinicians alike demand novel methods aimed to distill such complex data into simple—yet vibrant and behaviorally relevant—representations that can be interactively explored to discover new aspects of the data. Ideally, such representations could also be quantified to allow statistical inferences and provide the basis of future biomarkers for mental disorders. With these goals in sight, and while addressing several methodological gaps, we present a novel approach using TDA to examine the overall temporal arrangement of whole-brain activation maps. Without arbitrarily collapsing data in space or time, our TDA-based approach generates graphical representations of how the brain navigates through different functional configurations during a scanning session—i.e., a data-driven representation of the stream of mind that unfolds as participants lie in the scanner. These representations, when computed on a continuous multitask dataset, revealed the temporal arrangement of whole-brain activation maps as a hybrid of two mesoscale structures, i.e., community and core−periphery organization. Remarkably, the community structure was found to be essential for the overall task performance, while the core−periphery arrangement revealed that brain activity patterns during evoked tasks were aggregated as a core while patterns during resting state were located in the periphery. Neurophysiologically, the core represented task-specific (task-positive) brain activations, while the periphery represented task-unrelated (task-negative) activations. This neurophysiological insight indicates higher similarity of whole-brain activation patterns when participants are actively engaged in cognitively demanding tasks compared to when allowed to freely mind-wander during rest periods, which is a well-established hallmark of brain dynamics[21,50]. Lastly, by projecting shape graphs into the time domain, we were able to pinpoint between- as well as within-task transitions at the temporal resolution of a few time frames.

To fully appreciate the potential of our approach, it is important to compare it with previously devised methods for studying brain dynamics. As adequately stated by Preti et al.[1], the currently used methods fall primarily into two categories: those that conceptualize spontaneous brain activity as having a slow, yet continuously evolving, temporal dynamic; and others that suggest that all relevant information can be condensed into a sparse set of short events. As a method that relies on the whole-brain multivariate patterns of BOLD signal intensity as its input, our method lies conceptually closer to those methods that explore brain dynamics on the basis of sparse events (e.g., CAPs[21], paradigm free mapping[23], point process analysis[14]) rather than those that rely on the estimation of inter-regional (or inter-voxel) cofluctuations over time (e.g., sliding window Pearson's correlation[10], dynamical conditional correlation[51], multiplication of temporal derivatives[52]). At the same time, our method also distinguishes itself from the category of exploring dynamics based on sparse events, because shape graphs do not necessarily assume that brain dynamics arise from only a subset of significant events but permits exploration of the continuous unfolding of dynamics across each time frame. Further, unlike most previous approaches for studying brain dynamics, we do not require estimation of correlation (or connectivity) between parcellated brain regions and instead use whole-brain voxel-level activation maps to reveal the overall shape of brain dynamics. Taken together, our method may provide a bridge to gap the conceptual differences between the two prominent approaches for fMRI brain dynamics, or even provide a novel avenue to explore how to conceptualize brain dynamics best.

Using task-based fMRI data, our method could reveal transitions in whole-brain activity patterns at a temporal scale that is perhaps hitting the lower limit of the hemodynamics, due to the inherent delay in the hemodynamic response. Examination of brain dynamics in fMRI data at such a fine temporal resolution is currently not possible with sliding window-based methods. It can be argued that our method could be used in the future to better merge complementary neuroimaging modalities (e.g., electroencephalogram and fMRI recordings) for examination of brain dynamics at the highest possible spatiotemporal resolution. Further, by generating a representation of how the brain navigates through different functional configurations (or brain activation maps) at the level of individual time frames, our approach provides a handle representing the stream of mind that unfolds as the participant lies (or performs) in the MR scanner. We argue that this level of detailed representation of transitions in brain configurations makes our approach potentially useful in studying brain dynamics in general and developing biomarkers in mental illnesses in particular.

Our approach also provided novel biological insights, which were later validated using additional fMRI analysis. For example, participants with a cohesive community structure in their shape graph showed higher performance across different tasks. High modularity suggests that nodes are more likely to be connected to other nodes containing the same task type as compared to other task types. Thus, suggesting that participants who performed better across tasks had reliably evoked whole-brain configurations that were specific to each task, whereas participants who performed poorly had evoked whole-brain configurations that were more similar between (rather than within) task types. We tested

this insight by dividing the data into high- versus low-performance groups and analyzing similarity between whole-brain FC configurations across tasks blocks. As predicted, the participants in the high-performance group (compared to the low-performance group) had significantly lower similarity between FC configurations derived from different task blocks.

Another biological insight provided by our approach was related to the presence of both community and core–periphery mesoscale arrangements at the level of whole-brain activity maps. We also assessed the neurophysiological basis of such mesoscale arrangements. To compare and analyze different complex networks, estimation of these mesoscale structures is increasingly preferred as it can provide information regarding the overall architectural arrangement of a network[39]. In neuroscience, both of these mesoscale structures have recently been used to characterize how sets of brain regions segregate and integrate during complex cognitive processing. Using non-overlapping sliding windows of 2–3 min, Bassett and colleagues estimated the temporal variability in the region-by-region community structure during a motor learning task[38]. During this motor learning paradigm, brain regions were identified as part of a core if their community assignment did not change over time. The regions that were identified as part of the core included motor-related (left-lateralized primary sensorimotor) and visual processing brain areas, thus, suggesting that the core regions were primarily related to the task (i.e., motor learning) itself. Although our characterization of mesoscale properties aims to characterize the overall arrangement of brain activity patterns and not how individual brain regions interact with each other, it is nonetheless noteworthy to find similarity with previous work regarding the neurophysiological basis of high coreness, which we also observed to be engagement of task-related brain regions.

Using a multitask paradigm we aimed at capturing the overall dynamical arrangement that could represent how task-specific brain configurations interact with configurations that are shared between tasks. The three cognitive tasks used in the paradigm—working memory, math, and video—are known to elicit relatively distinct evoked brain activation patterns; however, these tasks also share several key low-level visuospatial and motor processing functions. Additionally, transitioning from one task to the next requires a common cognitive construct of task switching, which plays a significant role in rapidly reconfiguring brain networks so that humans can efficiently implement a variety of tasks one after another[53,54]. Thus, using our approach we aimed to capture this interplay between task-specific, low-level processing, as well as switching-specific brain activation patterns. The clinical evidence suggests that disturbances in such interplay are linked to psychiatric disorders[55,56]. In the future, approaches like ours that provide a unique avenue to quantify this interplay at the level of single participants could be used for translational outcomes.

Future research should evaluate whether the topological characteristics of shape graphs—namely community and core–periphery structures—could also capture clinically relevant markers. For example, depression is typically characterized by frequent rumination on negative self-referential thoughts and hyperconnectivity of the default mode network[57]. We hypothesize that such excessively long and frequent rumination periods will produce shape graphs with significantly larger cores for depressed patients compared to healthy controls.

Here, for validation purposes, we used fMRI data collected from task-based experimental paradigms where transitions between tasks were experimentally constrained. However, in the future, we plan to test our method's utility in revealing similar transitions in intrinsic (or at rest) data and relating topological properties of shape graphs extracted from those resting data to clinical and behavioral markers. Another avenue for future

research could be to gain mechanistic insights into how the brain's dynamical landscape (and associated temporal transitions) is altered by changes in internal brain state, external modulation (e.g., neuromodulation), as well as clinical conditions.

Methodologically, future work is needed to reduce the required computational resources of our approach, e.g., for real-time applications like neurofeedback. A potential way to better utilize computational resources and improve the speed of processing is to perhaps use gray-matter voxels only or coarse-grained spatial resolution as an alternative to individual voxels (e.g., about 90K grayordinates[58] versus ~300K voxels). Lastly, here, we focused on the two most common mesoscale architectures to characterize shape graphs. However, in the future, other graph-theoretical measures (e.g., rich clubs, assortativity or betweenness centrality) could also be explored to better relate individual differences in shape graphs with behavioral and clinical markers.

Looking forward, it can be argued that some brain disorders might be better characterized by aberrant transitions between different cognitive processes. For example, rapid transitions between cognitive processes may characterize attention disorders, while fixed states may indicate internal rumination, typical of depressive disorders. For a successful translational application of capturing and quantifying brain dynamics, it is important to develop methods that have single-participant specificity, are robust for a variety of acquisition paradigms, and are reliable in the face of partial data or parameter perturbations. Thus, methods that are robust in capturing and quantifying transitions across mental processes may be a promising addition to developing novel diagnostics.

## Methods

**Dataset 1: continuous multitask paradigm.** In this paper, we used two already collected fMRI datasets. The first dataset was originally collected by Gonzalez-Castillo et al.[34] using a CMP. We gathered these data from the XNAT Central public repository (https://central.xnat.org; Project ID: FCStateClassif). Informed consent was obtained from all subjects. The local Institutional Review Board of the National Institute of Mental Health in Bethesda, MD reviewed and approved the CMP data collection. Stanford University Institutional Review Board (IRB) reviewed and approved the use of these data in the current study. Brief details about these data and preprocessing steps are provided below.

This dataset contained de-identified fMRI and behavioral data from 18 participants who completed the CMP experiments as part of the study[34]. Details about the experimental paradigm are presented elsewhere[34]. Briefly, during the CMP task, participants were scanned continuously for a 25 min and 24 s long session, while performing four different tasks. Each task was presented for two separate 3 min blocks, with each task block being preceded by a 12-s instruction period. The order of task blocks was randomized such that each task was always preceded and followed by a different task. Once randomized, the experimental design was kept same for all participants. The four tasks included in this paradigm were: (1) Rest, where participants were instructed to look at the + sign on the screen and let their mind wander; (2) Math, where participants were presented with simple arithmetic operations, involving three numbers between 1 and 10 and two operands (only addition and subtraction). Operations remained on the screen for 4 s, and a blank screen appeared for 1 s between successive trials. This timing resulted in a total of 36 operations per each 3-min block; (3) Working Memory, where participants were presented with a continuous sequence of individual geometric shapes (appeared every 3 s) and were instructed to press a button whenever the shape currently on the screen was the same as two shapes before in the sequence (2-back design); and (4) Video, where participants watched a video of a fish tank from a single stationary point of view with different types of fish swimming in and out. Participants were instructed to look for a red-crosshair and signal by pressing a button whether the crosshair appeared on a clown fish or any other type of fish. These targets appeared for 200 ms with a total of 16 targets during each of the 3-min blocks.

The fMRI data were acquired on a Siemens 7 Tesla MRI scanner equipped with a 32-element receive coil (Nova Medical). Functional runs were obtained using a gradient recalled, single shot, echo planar imaging (gre-EPI) sequence {repetition time [TR] = 1.5 s, echo time [TE] = 25 ms, and voxel size = isotropic 2 mm}. A total of 1017 time frames were acquired in each session.

The behavioral data included responses and reaction times for the math, working memory, and video tasks. Participants were instructed to respond as quickly and accurately as possible with only one response per question. For each of the three tasks (math, memory, and video), percent correct, percent missed, and

reaction times were calculated. Both average and individual trial data were made available in the XNAT Central repository[34].

We performed standard fMRI preprocessing steps using the Configurable Pipeline for the Analysis of Connectomes (C-PAC version 0.3.4; http://fcp-indi.github.io/docs/user/index.html). Individual functional and anatomical MR images were transformed to a common 152 brain template that is maintained by the Montreal Neurological Institute (MNI)[59]. We used the Advanced Normalization Tool (ANTS) for registering images, as it has been shown to outperform other methods[60]. Registration involves a three-step process—(1) individual participant's anatomical images are first transformed to match the common template; followed by (2) registering individual participant's functional data to own transformed anatomical image, and finally, (3) functional derivative files are transformed to the common template. The fMRI data preprocessing included slice timing correction, motion correction (using FSL MCFLIRT tool), skull stripping (using FSL BET tool), grand mean scaling, spatial smoothing (FWHM of 4 mm) and temporal band-pass filter ($0.009$ Hz $< f < 0.08$ Hz). Additionally, nuisance signal correction was done on the data by regressing out (1) linear and quadratic trends; (2) mean time-series from the white matter (WM) and the cerebrospinal fluid (CSF); (3) 24 motion parameters obtained by motion correction (the six motion parameters of the current volume and the preceding volume, plus each of these values squared.); and (4) signals extracted using the CompCor algorithm[61]. We used five components for CompCor-based nuisance regression. To extract mean time-series from the WM, gray matter, and CSF the anatomical MR data were automatically segmented using FMRIB's Automated Segmentation Tool (FSL FAST tool). As the last step, these data were brought to 3 mm MNI space and normalized (demeaned with unit variance) before running the Mapper.

**Dataset 2: HCP working-memory paradigm.** The second dataset was originally collected as part of the Human Connectome Project (HCP)[35] while participants performed working-memory tasks. We gathered these data from the HCP website (https://db.humanconnectome.org). Informed consent was obtained from all subjects. The HCP scanning protocol was approved by the local Institutional Review Board at Washington University in St. Louis. Stanford University Institutional Review Board (IRB) reviewed and approved the use of these data in the current study. Brief details about these data and preprocessing steps are provided below.

De-identified data were downloaded from the subset of 40 unrelated participants, while they performed the working memory task[35,62]. This subset includes data from 38 individuals (21 F; age range = 22–35 years). Details of the experiment are presented elsewhere[35]. Briefly, the N-back working memory paradigm was employed using two 5-min runs. Within each run, four different stimulus types (faces, places, tools, and body parts) are presented in separate blocks, for a total of eight blocks (two per stimulus type). Out of the eight blocks, four blocks use a two-back working memory task (i.e., respond target whenever the current stimulus is the same as the one two-back) and the other four blocks use a 0-back working memory task (i.e., a target cue is presented at the start of each block, and the person must respond target to any presentation of that stimulus during the block). A 2.5 s cue indicates the task type (and target for 0-back) at the start of the block. In addition to eight working memory blocks, each run also includes four fixations (or resting state) blocks. The duration of each working memory block was set to 25 s, while the duration for resting state blocks was set to 15 s each.

The fMRI data were acquired using whole-brain EPI sequences, with a 32-channel head coil on a modified 3 T Siemens Skyra. The acquisition parameters included were as follows: TR = 720 ms, TE = 33.1 ms, Voxel size = 2.0 mm isotropic. A multi-band acceleration factor of 8 was used to increase temporal resolution. Two runs of working memory experiment were acquired, one with a right-to-left and the other with a left-to-right phase encoding.

The behavioral data included both responses and reaction times for the working memory task. Participants were instructed to respond as quickly and accurately as possible with only one response per question. Individual trial data were gathered from the Human Connectome Project database.

We downloaded minimally preprocessed data from the Human Connectome Project database. Details about this minimal processing are provided elsewhere[62]. Briefly, the preprocessing steps included gradient unwarping, motion correction, fieldmap-based EPI distortion correction, brain-boundary-based registration of EPI to structural T1-weighted scan, non-linear (FNIRT) registration into MNI152 space, and grand-mean intensity normalization. Additionally, nuisance signal correction was done on the data by regressing out 12 motion parameters obtained by motion correction (the six motion parameters of the current volume and the preceding volume). Lastly, these data were brought to 3 mm MNI space, band-passed (0.009–0.08 Hz), normalized (demeaned with unit variance) and spatially smoothed (4 mm) before running Mapper.

**TDA-based Mapper analysis pipeline.** Although the details about the TDA-based Mapper approach are presented elsewhere[26,27,29], we provide a brief summary for each of the five Mapper steps. As a first Mapper step, without collapsing data in space or time, for each individual, preprocessed 4D fMRI data were transformed into a 2D matrix. The rows of this 2D matrix represented time frames whereas the columns represented voxels. For the CMP data, the size of each individual's 2D input matrix was [1017 × 271,633]. Similarly, for the HCP data, the size was [405 × 271,633].

As a second Mapper step, filtering was used for dimensionality reduction. A variety of filters can be used for this step, namely, geometric filters (e.g., distance-based density, measures of centrality, etc.) or non-geometric filters (e.g., derived from PCA or projection pursuit analysis). We used the Neighborhood Lens[63] function to project the high-dimensional data (in 271,633 dimensions) to two dimensions. This function is a nonlinear dimensionality reduction method that uses a variant of SNE (t-SNE[36,37]). Nonlinear methods like t-SNE allows for preservation of the "local" structure in the original high-dimensional space after projection into the low-dimensional space, which is typically not possible with linear methods like PCA or MDS[36].

As a third Mapper step, the filter range was divided into overlapping bins. The number of bins and amount of overlap is determined by resolution parameter ($R$) and gain parameter ($G$), respectively. For the CMP fMRI data, these parameters were selected as part of the data-driven automated approach (see Data-driven parameter search). This approach selected $R_{CMP} = 30$ and $G_{CMP} = 3$. Please also see the Parameter perturbation analysis section, where we showed the reliability of our results under a variety of different parameter values. For the replication HCP fMRI data, the filter function set and binning parameters were kept the same as that of CMP (with the only exception of resolution parameter ($R$), which was proportionally modified to match the higher sampling rate of the HCP data, i.e., instead of using $R_{CMP} = 30$, we used $R_{HCP} = R_{CMP} \times (TR_{HCP}/TR_{CMP}) \approx 14$).

As a fourth step, partial clustering is performed, within each bin, to reduce the complexity of the shape graph. The resulting clusters from this step later become nodes in the shape graph. The Mapper algorithm is not tied to any particular clustering approach. Here we used the single-linkage clustering algorithm, as it does not require specifying the number of clusters beforehand. Further, previous applications of Mapper to a variety of datasets have successfully used this approach[29]. The distance metric for single-linkage clustering could be chosen to be Euclidean or correlation or any other similarity function. We used the data-driven approach to find the best distance metric, which turned out to be Manhattan L1.

Finally, as a fifth step, a combinatorial object or shape graph was generated from the low-dimensional compressed representation. The Mapper treats each cluster as a node in the graph and connects these nodes with an edge if they share one or more time frames.

**Visualization of shape graphs.** The shape graphs were annotated (or colored) by task type. The nodes with time frames from multiple tasks were visualized using pie-charts to appropriately depict the proportion of time frames from each task. A web-based interface was also developed to interact with the shape graph (Supplementary Fig. 2). This implementation was developed using HTML5, Scalable Vector Graphics (SVG), CSS, and JavaScript. Specifically, we used the D3.js framework (Data-driven documentation; D3) for displaying individual participants' shape graphs and the associated spatial profiles (generated from the mixture modeling at each time frame). The interface also displays spatial correlation with known large-scale brain networks in real time.

The node annotation (color and pie charts) can be changed in real time to display other pieces of information. For example, by clicking Hit/Miss button in the web tool, the nodes can display the proportion of hit (or correct response) versus miss (or incorrect response) trials (Supplementary Fig. 3). Such information can be used in future to mine information about trial-to-trial variation in each participant.

Lastly, the tool allows for generating temporal movies to review how spatial topographies (and associated spatial correlation with large-scale brain networks) changes in time as the participant transitions from one task to the next. Supplementary Movie 1 provides an example for such temporal animations for one of the representative participants ($S_{01}$).

**Quantifying mesoscale structure of shape graphs.** To quantify the community structure in a shape graph, we estimated the widely used quality function $Q_{mod}$[64]. Mathematically, for a given graph $G$, with $N$ nodes and a set of edges $E$ connecting those nodes, $Q_{mod}$ can be defined as follows:

$$Q_{mod} = \sum_{\{i,j\}} [A_{ij} - P_{ij}] \delta(g_i, g_j),$$

where $A$ is the adjacency matrix, with $A_{ij}$ as cell elements containing the weight of connection between nodes $i$ and $j$. For a hard partition (i.e., where each node is assigned to exactly one community) and where $g_k$ denotes the community for node $k$, the function $\delta(g_i, g_j) = 1$ if $g_i = g_j$ and equals to 0 otherwise. $P_{ij}$ denotes expected connection strength between nodes $i$ and $j$, under a specified null model. One of the most common null model[64] is given by,

$$P_{ij} = \frac{k_i k_j}{2m},$$

where $k_i$ is the strength of node $i$, $k_j$ is the strength of node $j$, and $m = \frac{1}{2} \sum_{ij} A_{ij}$. The community assignment for each node in shape graph was chosen to be one of the four tasks (i.e., Rest, Working Memory, Video, and Math) based on the majority of time frames belonging to any one task.

To quantify the core−periphery structure in each participant's shape graph, we employed the generalized Borgatti and Everett[41] algorithm that provides a coreness score (CS) for each node along a continuous spectrum between nodes that lie most

deeply in a network core with a CS ~ 1 and those that are in the periphery with a CS ~ 0[38]. We used the implementation by Rombach et al.[39] to estimate CS, which was designed for undirected networks. This method takes into account cores of different shapes and sizes by giving credit to all nodes and by weighting the credit using a quality function $R_{(\alpha,\beta)}$, defined below,

$$R_{(\alpha,\beta)} = \sum_{ij} A_{ij} C_{ij},$$

where $(\alpha,\beta)$ are the two parameters of this approach, such that $\beta$ sets the size of the core and $\alpha$ sets the boundary between core and periphery. A large value of $\alpha$ indicates sharp transition. The symbol $A$ denotes the adjacency matrix, with $A_{ij}$ as cell elements containing the weight of connection between nodes $i$ and $j$. The elements $C_{ij}$ of the core matrix are given by $C_{ij} = C_i C_j$, and $C_i \geq 0$ is the local core value of node $i$. The local core values of node $i$, $C_i$, is estimated by maximizing $R_{\alpha,\beta}$ using Simulated Annealing (as implemented in Matlab®). The aggregate of coreness score of each node $i$ is,

$$CS(i) = Z \sum_{(\alpha,\beta)} C_i(\alpha,\beta) \times R(\alpha,\beta),$$

where $Z$ is a normalization factor such that the CS(i) normalizes to a maximum value of 1. As there is no a priori on how to choose $(\alpha,\beta)$ values, we sampled $\alpha$ and $\beta$ uniformly over a discretization of the square [0,1]×[0,1] and reported the averaged CS(i) for each node $i$. This uniform sampling approach to choose $(\alpha,\beta)$ has been prescribed previously[39].

**Extracting temporal transitions.** To quantify temporal transitions associated with the task-related brain activity, we simply estimated the degree of nodes in the TCM. The degree for each node (or time frame) in the TCM was estimated by counting the number of non-zero edges connected to that time frame.

**Data-driven parameter search.** To find the best set of TDA-based Mapper parameters (e.g., Gain and Resolution parameter), we used the data-driven outcome-auto analysis feature, as implemented in the software API[63]. In this analysis, the parameter search for TDA-based Mapper approach is done to optimize localization of a given feature in the shape graph. For example, given a sample dataset containing a part of the observations from male participants and the other part of female participants, the outcome-auto analysis could then be used to find TDA parameters that best localizes data from the same sex participants nearby in the shape graph. In our case, using outcome-auto analysis, we optimized TDA parameters to localize the task type (i.e., rest, working memory, etc.). This optimization was done only for the CMP dataset. To test the reliability of these parameters, we later successfully used the set of parameters optimized on the CMP data for the replication HCP data.

The details regarding this optimization are presented elsewhere[63]. Briefly, the algorithm uses a two-pass approach—where it first searches for the set of filter functions and distance metric that best localizes some outcome measure (here we used task type as an outcome measure) and in the second pass, for a given set of filter functions and distance metric, the algorithm searches for a set of binning parameters that best localize the outcome measure on the shape graph. Table 1 provides the final parameters.

**Parameter perturbation analysis.** Although a data-driven approach was used to objectively find parameter values for generating shape graphs, as an additional measure towards estimating the reliability of shape graphs, we tested the effect of parameter perturbation on shape properties and their relation to behavioral task performance. For the perturbation analysis, we widely varied the two main TDA binning parameters—i.e., the number of bins (or resolution, $R$) and percentage of overlap between bins (or gain, $G$)—to generate 49 different variations of the shape graph for each CMP participant. These two binning parameters largely control the overall arrangement of shape graph. Thus, to test whether the shape graph properties (e.g., core−periphery arrangement) is robust in the face of perturbing binning parameters, we varied $R$ from 10 to 70 (steps of 10) while $G$ was varied from 2 to 5 (steps of 0.5). Results are shown in Supplementary Figs. 8 and 10. Overall the shape properties (e.g., the core−periphery structure) were reliably observed in a majority of parameter variations. Further, the association between modularity and

task performance was also significant and reliable across a majority of parameter variations.

**Traditional GLM analysis.** The aim of these analyses was to reveal brain regions that were positively (or negatively) associated with CS and hence provide neurophysiological basis for the core (or periphery) nodes in the shape graph. A weighted GLM analysis was performed. First, the coreness score of each node was mapped back to the time frames contained in that node. Thus, if a node has a CS of 0.5, then the corresponding time frames contained in that node also got a proportional weighting of 0.5 in the GLM analysis. Using a multiple regression analysis, weighted time frames were entered for each task separately creating four explanatory variables. Two contrasts were run to examine brain regions that show positive as well as negative association with the coreness scores. We qualitatively compared the cluster-corrected ($Z > 2.3$ and FWER $p < 0.05$) group-level results with task-specific meta-analysis statistical maps derived from the NeuroSynth library[44].

**Whole-brain functional connectivity configurations.** To examine whether the high-performance group elicited task-specific brain configurations, we estimated whole-brain FC configurations during each task block for each participant. To estimate these brain configurations—first, data were sampled from a set of 264 brain regions to make inferences at the regional and systems level (Supplementary Fig. 12A). We used an independent and well-established brain parcellation scheme that was previously identified using a combination of resting-state FC parcellation as well as task neuroimaging meta-analysis[45]. Data were summarized for each region by averaging signal in all voxels falling inside a sphere (radius = 5 mm) centered at the coordinates provided by Power et al.[45]. The FC between each pair of brain regions was estimated using Pearson correlations ($r$). These correlation values were converted to Fisher's $z$-transform for further analysis. After estimating task-based FC matrices, we calculated similarity between FC matrices across different task blocks within each participant. For a quantitative group-level comparison, the average similarity between FC matrices for each participant was estimated (average over upper triangle of the participant's similarity matrix) and a two-sample $t$ test was run to compare the low-performance group from the high-performance group.

**Null model generated using neuRosim.** To benchmark and validate the results generated from our approach, we employed three different null models. The first null model was designed to test whether the metrics generated from our approach, including the shape graph and its characteristics, are mainly driven by physiological signals (e.g., cardiac and respiratory) and spatiotemporal noise. In addition to these main sources of noise, the individual variations in neuroanatomy were also included in the model, by using each individual's anatomical scan (T1-weighted scan) as a baseline. The 4D fMRI null datasets were simulated using an R-package (neuRosim[65]). To better model the spatiotemporal properties of real 4D fMRI data, the null model includes four different noise sources: (a) white system noise ("rician" noise); (b) physiological noise (1.17 Hz for heart beat and 0.2 Hz for respiratory rate); (c) temporal noise (using auto-regressive modeling with model order chosen using the AIC); and (d) spatial noise (using Gaussian Random Field with FWHM = 4 mm). Lastly, each of the noise parameters was modeled independently for the gray matter, WM, and CSF. These null models were generated for each individual. The autocorrelation function (ACF) of null data were observed to follow the ACF of real data very closely (see Supplementary Fig. 11).

**Null model generated using phase-randomization.** The second and third null models were designed to test whether the metrics and shape graph generated by our approach is mainly driven by task-unrelated non-stationarity in the real data. To generate these null models, phase randomization of the original dataset was done independently for each individual. Phase randomization involves randomizing the observed time series by performing Fourier transform, scrambling the phase and then inverting the transform to get the null model[66]. The autocorrelation function, power spectrum, and other linear properties are preserved under phase randomization[67]; also see Supplementary Fig. 11. Further, the two variants in the phase randomization procedure include scrambling the phase using the same sequence across all time-series versus scrambling the phase using a random sequence for each time-series. The former approach preserves covariance structure in the data while randomizing nonlinear properties in the data. However, the latter also disrupts linear relationships in the data. We used both of these approaches here to test the validity of our methods. Matlab scripts were used to generate phase randomization-based simulations.

**Code availability.** To generate shape graphs, we used a licensed version of TDA software through the Ayasdi cloud-based platform (www.ayasdi.com). Open source versions of the TDA code are also available in Python (http://danifold.net/mapper/introduction.html) and R (https://cran.rproject.org/web/packages/TDA/index.html). To visualize data, an in-house web-based interface was developed. Matlab scripts were written to analyze the shape graphs; this code is available from the authors upon reasonable request.

**Table 1 TDA parameters revealed by a data-driven approach applied on the CMP dataset**

| TDA parameter | Value is chosen based on outcome-auto analysis |
|---|---|
| Distance metric | Manhattan (L1) |
| Filter function set | Neighborhood Lens 1 and 2 |
| Resolution parameter | 30 |
| Gain parameter | 3 |

**Data availability**. The CMP data used in this work were originally collected by Gonzalez-Castillo et al.[34]. We gathered these data from the XNAT Central public repository (https://central.xnat.org; Project ID: FCStateClassif). The second dataset was originally collected as part of the Human Connectome Project (HCP)[35] while participants performed working-memory tasks. We gathered these data from the HCP website (https://db.humanconnectome.org).

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

## Acknowledgements

Funding was provided by a Career Development Award (K99MH104605) from the National Institute of Mental Health (NIMH) and an NARSAD Young Investigator Grant from the Brain & Behavior Research Foundation to M.S. Funding for the CMP data were provided by the National Institute of Mental Health Intramural Research Program (NIH clinical protocol number NCT00001360, protocol ID 93-M-0170, annual report ZIAMH002783-16; Principal Investigator: P.A.B.). Funding for the HCP data were provided by the 16 NIH Institutes and Centers that support the NIH Blueprint for Neuroscience Research (as part of the Human Connectome Project, WU-Minn Consortium; Principal Investigators: David Van Essen and Kamil Ugurbil; 1U54MH091657) and by the McDonnell Center for Systems Neuroscience at Washington University. The authors thank the staff at Ayasdi Inc. for their computational support and guidance.

## Author contributions

M.S. designed the study, developed methods, performed analysis and wrote the manuscript. O.S., G.G., G.C., J.G.-C., P.A.B., and A.L.R. contributed to study design, interpreting results and writing the manuscript. G.C. also helped with developing methods specific to TDA. O.S. also helped with developing methods specific to graph theory.

## Additional information

**Competing interests:** G.C. is an employee of Ayasdi Inc. and owns shares of the company. The remaining authors declare no competing interests.

