## [Peer Review File(PDF 4296 kb) · Nature Communications]

Editorial Note: this manuscript has been previously reviewed at another journal that is not operating a transparent peer review scheme. This document only contains reviewer comments and rebuttal letters for versions considered at *Nature Communications*.

Reviewers' comments:

Reviewer #1 (Remarks to the Author):

The authors' revised manuscript covers all points raised during the previous round of review; however, the arguments put forth in the rebuttal are mostly based on speculation and do not offer concrete evidence in support.

For example, in response 2.1 the authors assert that "participants who performed better across tasks had reliably evoked brain activity patterns that were specific to each task"; however, a straightforward control analysis could have corroborated this result from the shape graphs with an independent calculation. Further, the authors' response to Comment 3 provides mostly speculative arguments for the mixture of tasks in the tendrils, which does not address the fundamentally confusing point: if the authors claim that their approach reveals a unique distinction between activity during resting state and task conditions, does the mixture of tasks in the tendrils also imply something about brain activity during these states? As a simple example, should we conclude that a mixture of three tasks in a tendril ("Instructions", "Video", "Math") actually implies these activity patterns are similar (which remains unexplained), or is this just an artifact of the analysis method (which has not been ruled out by the control analyses)? If this point is an artifact of the analysis, effectively overfitting in the data clustering, it casts some doubt on the clustering of resting-state nodes into distinct peripheral tendrils.

For these reasons, it remains unclear whether the submission and its introduced techniques provide unique biological insight beyond previous work. The submission lacks the technical rigor and demonstrated novelty to appeal to the broad readership of the Journal. I do believe these results are interesting and have potential but would be more suited to a specialty journal.

Reviewer #3 (Remarks to the Author):

The authors have addressed my concerns. I congratulate them on a novel and important contribution to the literature.

Response to reviewer letter

Reviewer #1 (Remarks to the Author):

Comment #1: The authors' revised manuscript covers all points raised during the previous round of review; however, the arguments put forth in the rebuttal are mostly based on speculation and do not offer concrete evidence in support.

Response: We thank the reviewer for evaluating the revised manuscript. We respectfully disagree with the generic 'speculation' comment. Below, we have provided a point by point rebuttal to the issues raised by the reviewer. We have also included a new control (or validation) analysis. We have accordingly revised the manuscript and highlighted the changes in blue color font.

Comment #2: For example, in response 2.1 the authors assert that "participants who performed better across tasks had reliably evoked brain activity patterns that were specific to each task"; however, a straightforward control analysis could have corroborated this result from the shape graphs with an independent calculation.

Response: The assertion that "participants who performed better across tasks had reliable and task-specific activation" is not a speculation. Instead, it is based on the quantitative graph theoretical results. Further, we have now added a new control analysis to validate this assertion (see revised manuscript lines: 310-327, 561-565 and Supplementary Figure S12, which is pasted below for ease).

First, our assertion is based on the quantitative graph theoretical results. As reported in the Results section 2.2 and Figure 3 (pasted below for ease), task performance was observed to be directly associated with the modularity structure of the derived shape graph, such that higher performance on the task was observed in individuals with "high modularity" shape graphs. Modularity is a mathematical measure (defined Online Methods Section B.7) that provides an index of how segregated versus integrated a graph structure is. As a reminder, a TDA-derived shape graph represents how the whole-brain activation patterns at each time point (or acquisition sample) are related to activation patterns at every other time point during an MR-scan session. Thus, higher modularity in a shape graph suggests that nodes (representing time points) are more likely to be connected to other nodes containing the *same task-type* as compared to other task-types. In other words, participants who performed better across tasks had evoked brain activity patterns that were specific to each task (and hence higher modularity). One such example is the graph for subject S07 in the figure reproduced below (Figure 3A – right graph). Similarly, participants who performed poorly on the tasks had evoked brain activity patterns that were less specific to each task and more similar between task-types (i.e., lower modularity). An example of a low modularity graph for a bad performing subject (S14) is shown on the left on the figure below (Figure 3A – left graph).

Second, a new control analysis has now been added to the revised manuscript (see lines: 310-327, 561-565 and supplementary figure S12). Here, an independent whole-brain functional connectivity (FC) analysis was run to validate the TDA-derived finding that individuals with higher task performance putatively invoked task-specific brain configurations as compared to individuals with lower task performance. To do this,

we first performed a median split based on the overall task performance to divide the participants into two groups (low (n=9) and high performers (n=9) with a median split at 86.9% accuracy). Based on our TDA-derived shape graph analysis, we hypothesized that participants in the high-performance group would elicit task-specific brain configurations for each task block (duration 3 min) with relatively less similarity between brain configurations from different tasks. Similarly, for the low-performance group, we hypothesized that the brain configurations during each task block would be relatively more similar to different task blocks.

To examine these hypotheses, we first estimated brain configurations during each task block for each participant. Given the long duration (3 min) for each task block, only two repetitions per task, and low statistical power (n=9 per group) for traditional univariate analysis, whole-brain functional connectivity (FC) (between each pair of brain regions) was estimated as a proxy for brain activation pattern for each task block. Previous work has shown that whole-brain FC matrices reliably encode task-specific brain activation patterns^{1,2}. To estimate these brain configurations – first, data were sampled from a set of 264 brain regions to make inferences at the regional and systems level (Supplementary Figure 12A). We used an independent and well-established brain parcellation scheme that was previously identified using a combination of resting-state functional connectivity parcellation as well as task neuroimaging meta-analysis³. Data were summarized for each region by averaging signal in all voxels falling inside a sphere (radius = 5mm) centered at the coordinates provided by Power et al. (2011)³. The FC between each pair of brain regions was estimated using Pearson correlations (r). These correlation values were converted to Fisher's z-transform for further analysis. The whole-brain FC matrices for each task block are shown in Supplementary Figure 12B for a representative participant (S01). After estimating whole-brain FC matrices for each task block, we calculated the similarity between FC matrices derived from different task blocks within each participant. Supplementary Figure 12C shows the similarity between FC matrices derived different tasks for two representative participants (S14 – from low-performance group and S07 – from high-performance group). As descriptively evident, FC matrices from S14 were more similar between tasks as compared to FC matrices from S07. For a quantitative group-level comparison, the average similarity between FC matrices for each participant was estimated (average over the upper triangle of the between-task similarity matrix), and a two-sample t-test was run to compare the low-performance group from the high-performance group. A significant t-statistic ($t(16) = 2.50, p=0.0236$) was observed, such that participants in the low-performance group (compared to the high-performance group) had higher average similarity between whole-brain FC matrices derived from different task blocks. Thus, validating the TDA-derived assertion that participants who performed better across tasks evoked task-specific brain configurations.

Taken together, both the theoretical and the new experimental data provide support towards the TDA-derived prediction that a participant with higher task-performance evoked task-specific brain configurations as compared to participants with lower task-performance.

Figure 3: Quantifying community structure of shape graphs. (A) Shows two shape graphs from two representative participants (S14 and S07). As depicted, S14 had a low modularity score, i.e., nodes with different task types are connected to each other without any preference for same task nodes. On the other hand, shape graph from S07 depicts high modularity structure with nodes preferentially connected to other nodes of same task type. (B) The modularity score (Q_{mod}) was observed to be associated with task performance, such that higher modularity in the shape graph was associated with better task performance.

Supplementary Figure S12: Validation of TDA-derived prediction that participants with higher task-performance evoked task-specific brain configurations as compared to participants with lower task-performance. Using the overall task performance, a median split was used to divide the participants into two groups (low- ($n=9$) and high-performers ($n=9$) with a median split at 86.9% accuracy). An independent and well-established brain parcellation scheme³ was used to parcellate the brain into 264 brain regions (as shown in A). (B) Shows the whole-brain functional connectivity (FC) matrices for each task block for a representative participant (S01). (C) Shows similarity between FC matrices derived different tasks for two representative participants (S14 – from the low-performance group and S07 – from the high-performance group). Please note that that the obvious value ($=1$) of diagonal elements was zeroed-out. As descriptively evident, FC matrices from S14 were more similar between tasks as compared to FC matrices from S07. (D) Shows quantitative group-level comparison – average similarity between FC matrices for each participant was estimated and a two-sample t-test was run to compare the low-performance group from the high-performance group. A significant t-statistic ($t(16) = 2.50$, $p=0.0236$) was observed, such that participants in the low-performance group (compared to high-performance group) had higher average similarity between FC matrices derived from different task blocks.

Comment #3: Further, the authors' response to Comment 3 provides mostly speculative arguments for the mixture of tasks in the tendrils, which does not address the fundamentally confusing point: if the authors claim that their approach reveals a unique distinction between activity during resting state and task conditions, does the mixture of tasks in the tendrils also imply something about brain activity during these states? As a simple example, should we conclude that a mixture of three tasks in a tendril ("Instructions", "Video", "Math") actually implies these activity patterns are similar (which remains unexplained), or is this just an artifact of the analysis method (which has not been ruled out by the control analyses)? If this point is an artifact of the analysis, effectively overfitting in the data clustering, it casts some doubt on the clustering of resting-state nodes into distinct peripheral tendrils.

Response: We apologize for any lack of clarity. We do not claim that our method reveals a unique distinction between activity during rest and task conditions. Instead, our observation is that, on average, brain activation maps during “resting” state were represented in the peripheries of the shape graphs, while task-related brain activation was represented inside or around the core (quantified in the Results Section 2.2). As reported in the manuscript, our *main aim* is to reveal the overall organization of whole-brain activity maps during a multi-task (or multi-condition) experimental design without arbitrarily collapsing data in space or time.

As mentioned in our earlier response and revised text in the manuscript (lines: 509-520), the reason for why tasks other than resting state were represented on the peripheries (or tendrils) is mainly due to the efficacy of our approach in representing transitions in functional configurations (or brain activation maps) at the level of single time frames (unlike previous work where temporal windows of 30 s or more are typically used). Thus, we claim that our approach generates a putatively data-driven representation of the stream of mind that unfolds as participants lie in the scanner.

To make sure that the core-periphery arrangement observed in the shape graphs is not an artifact of the analytical approach, we ran several control analyses (already included in the originally submitted manuscript) using three different null models (see Results section 2.6 and Supplementary Figure S9). As shown in the manuscript, the shape graphs derived from the three null models do not have any core-periphery arrangement, indicating that the presence of such graph motifs is not an artifact of the analysis.

Furthermore, to check whether the observed core-periphery arrangement has a neurophysiological basis we ran weighted GLM (see Results section 2.3, Figure 4C and Supplementary figure S5). We searched for brain regions that positively (or negatively) correspond with increasing coreness scores. As shown in Figure 4C (pasted below for ease), higher coreness scores were associated with higher engagement of task-specific brain regions. Further, as shown in Supplementary Figure S5 (pasted below for ease), lower coreness scores (indicating peripheral nodes or nodes on the tendril) were associated with activation in the task-nonspecific default mode regions of the brain (especially the Posterior Cingulate Cortex) – typically associated with mind wandering.

Based on the aforementioned findings, we argued first that the results are not an analytical artifact; and second that the presence of task-related nodes in periphery (instead of densely connected core) could be generated either by transitions between time frames (or nodes) from one task to the next or by lapses in attention during task blocks (e.g., due to mind wandering). Such invariable lapses in attention are especially possible in task blocks where low cognitive engagement is required. For example, due to the experimental design, the cognitive engagement required for each task varied from the lowest (resting state and instruction condition) to medium (video task) to highest engagement (working memory and math).

Altogether, we argue that this level of detailed representation of transitions in brain configurations is what makes our approach unique and useful for studying brain dynamics, as well as potentially a useful tool for the development of biomarkers for mental illnesses.

Figure 4(C) Shows the brain regions that were positively associated with coreness scores, i.e., higher the coreness score (CS), higher the activation in these regions. The group-maps (in red-yellow scale) for positive relation with CS are overlaid on the task-associated meta-analysis maps from NeuroSynth⁴. As shown, higher CS was associated with higher engagement of task-related brain regions.

Supplementary Figure S5: Shows the significant brain activation maps for brain regions that were negatively associated with the coreness scores (CS), within each cognitive task. To better depict the overlap across the three tasks, the maps are thresholded and binarized at cluster correction FWER $p < 0.05$ using $Z > 2.3$. Across the three cognitive tasks, we found a negative association with CS in the posterior cingulate cortex region (PCC). This overlap suggests that lower CS (or a peripheral node) was associated with task-nonspecific brain activation in the regions usually linked with intrinsic processing at rest (or default mode network).

Comment #4: For these reasons, it remains unclear whether the submission and its introduced techniques provide unique biological insight beyond previous work. The submission lacks the technical rigor and demonstrated novelty to appeal to the broad readership of the Journal. I do believe these results are interesting and have potential but would be more suited to a specialty journal.

Response: We respectfully disagree with this assessment of our work. The revised manuscript not only provides evidence for both functional and biological significance of our work but also was conducted with extreme technical rigor. We performed thorough reliability and validation analysis for the proposed approach,

including (1) comparison with three different null models, (2) replicating our results in an entirely independent data set from the Human Connectome Project (HCP), (3) a systematic parameter perturbation analysis, and (4) anchoring the observed properties into neurophysiology.

References:

1. Gonzalez-Castillo, J. *et al.* Tracking ongoing cognition in individuals using brief, whole-brain functional connectivity patterns. *Proc. Natl. Acad. Sci. U.S.A.* **112**, 8762–8767 (2015).
2. Finn, E. S. *et al.* Functional connectome fingerprinting: identifying individuals using patterns of brain connectivity. *Nat. Neurosci.* **18**, 1664–1671 (2015).
3. Power, J. D. *et al.* Functional network organization of the human brain. *Neuron* **72**, 665–678 (2011).
4. Yarkoni, T., Poldrack, R. A., Nichols, T. E., Van Essen, D. C. & Wager, T. D. Large-scale automated synthesis of human functional neuroimaging data. *Nat Methods* **8**, 665–670 (2011).

REVIEWERS' COMMENTS:

Reviewer commenting on your response to the previous Reviewer 1's concerns:

All concerns have been addressed.